# Experimental Investigation of Special-Shaped Concrete-Filled Square Steel Tube Composite Columns with Steel Hoops under Axial Loads

**DOI:** 10.3390/ma15124179

**Published:** 2022-06-13

**Authors:** Zhen Wang, Zhe Liu, Xuejun Zhou

**Affiliations:** 1School of Civil Engineering, Shandong Jianzhu University, Jinan 250101, China; wangzhen_sdjzu@163.com; 2Shandong Winbond Construction Group Co., Ltd., Weifang 262500, China

**Keywords:** special-shaped concrete-filled steel tube, axial load, load-carrying capacity, bearing mechanism, failure mode, finite element analysis

## Abstract

Special-shaped concrete-filled steel tube (SS-CFST) columns can be embedded in the wall, thus preventing the columns from protruding. This feature makes it popular in steel residential buildings. This paper proposes a new special-shaped concrete-filled square steel tube (SS-CFSST) composite column composed of multiple square steel tubes connected by steel hoops to form L-, T- or cross-shaped sections. Eight specimens were tested under axial loads with section shape, construction method, slenderness ratio, steel tube thickness, and steel strength as variation parameters. The structural performance, such as failure modes, peak load, load–displacement curves, load–strain curves, and Poisson’s ratio of the steel tubes, were analyzed. The tests illustrated that the failure modes of hoop-type specimens and weld-type stub columns were mainly the local buckling of steel tubes and bending failure, and those of the weld-type slender columns were mainly overall bending failure. The load-carrying capacity of the hoop-type specimen was higher than that of the weld-type specimen with the same cross-sectional dimensions and slenderness ratio. Next, the stress–strain relationship model of core concrete in the SS-CFSST composite column was established by considering the restraint effect of the connection coincidence area of steel tubes and steel hoops on concrete. Additionally, the finite element model (FEM) of the column was established using this constitutive model. By comparing the failure modes, load–strain curves and bearing capacities obtained from the tests and FEM, the established FEM can accurately evaluate the mechanical properties of SS-CFSST composite columns with steel hoops under axial compression.

## 1. Introduction

In recent years, prefabricated assembly construction has become a meaningful way to industrialize construction in the face of labor shortages and an aging workforce. The concrete-filled steel tube (CFST) structure, as an important structural form of prefabricated buildings, offers the advantages of fast construction, high bearing capacity, good seismic performance, and no need for additional formwork [1,2]. Han [3] and Uy [4,5] have undertaken various experiments and analyses on the circle and rectangular CFST column. It was found that the carrying capacity of the CFST column is higher than the sum of steel’s and concrete’s bearing capacity due to the restraint of steel tube on concrete and the supporting effect of concrete on steel tube. The CFST column is more economical in practical engineering. The special-shaped concrete-filled steel tube (SS-CFST) composite column combines a CFST structure with a special-shaped column. This combination solves the problem of columns protruding from walls in residential buildings, affecting furniture placement. It is a structural form suitable for prefabricated steel residential buildings.

The traditional method of constructing SS-CFST composite columns is to weld several rectangular steel plates into a specifically shaped tube and then pour concrete into the tubes. The section diagram of the traditional SS-CFST column is shown in Figure 1a–c. The relevant studies conducted by Chen [6] and Shen [7] found that the traditional SS-CFST composite columns exhibit excellent mechanical properties. However, there are also problems, including the special-shaped steel tube being prone to local buckling and the core concrete being weakly restrained in concave corners. A variety of construction methods have been proposed to improve the special-shaped steel tube’s restraint effects on the concrete and delay the local buckling of the steel tube.

Zuo [8] proposed SS-CFST composite columns with binding bars and compared the impact of binding bars on the columns’ damage modes and ultimate bearing capacities under axial compression. Horizontal bolt bars are set along the height direction on the special-shaped steel tubes, as presented in Figure 1d–f. The research indicated the setting of the binding bars delays the local buckling and contributes to the improved bearing capacity and ductility. Yang [9] carried out a series of mechanical tests on SS-CFST composite columns with stiffeners, as illustrated in Figure 1g–i. The stiffeners have little impact on the bearing capacity but enhance the contact between the steel tube and the concrete and eliminate the steel tube and concrete separation at the concave corner. A multi-cell CFST composite column is a structural form of several rectangular steel tubes combined by welding seams. Its cross-sectional shapes include I, L, and T. Song [10], Sui [11], Cui [12], and Sun [13] completed a series of tests to study mechanical performance under compression and seismic behavior under low cyclic loading of multi-cell CFST composite columns. The multi-cell CFST composite column enhances the restraining effects of steel on concrete and accentuates the combined performance of both materials.

The structures mentioned above can effectively improve the steel tube’s restraint effects on concrete, thereby postponing the steel tube buckling. However, there are other problems, such as high construction accuracy requirements for binding bars, large welding workload, and thick steel plate bending difficulties. To address these problems, Wang [14] proposed a weld-type special-shaped concrete-filled square steel tube (SS-CFSST) column, which is made up of several square steel tubes connected by welding seams at the chamfers. As the aspect ratio of the column’s cross-section is usually not greater than three, the L-shaped CFSST composite column consists of three steel tubes with a core column in the middle position and two limb columns on both sides. Similarly, the T-shaped section consists of four steel tubes, and the cross-shaped section consists of five steel tubes. The section diagram of the weld-type SS-CFSST composite column is presented in Figure 2a–c. The weld-type SS-CFSST composite column solves the stress concentration problem at the concave corner of the special-shaped section. The square section has stronger restraint on core concrete than the rectangular section, and the CFSST column has a higher bearing capacity and ductility. The weld-type SS-CFSST composite columns have been used in the Jianing Community, a public rental housing project. However, in engineering practice, the welding process along the entire column height causes large residual deformation of the steel tube, which will affect the processing and production accuracy.

This research proposes an SS-CFSST composite column with steel hoops, as illustrated in Figure 2d–f. Steel hoops connect the square steel tubes instead of welding seams. Eight specimens were tested under axial load to estimate the structural performances of the SS-CFSST composite column with steel hoops. The test phenomena, failure modes, load carrying capacities, and bearing mechanisms of the SS-CFSST composite column with steel hoops and weld-type specimens were compared. Based on the test results, the confined concrete’s stress–strain relationship of the SS-CFSST composite column with steel hoops was established, and a finite element model (FEM) was established to verify the accuracy.

## 2. Experimental Program

### 2.1. Specimen Design and Processing

Eight specimens were designed for this test, four with steel hoops and four with welding. The main parameters included section shape, construction method, slenderness ratio, and steel tube thickness. In accordance with the loading capacity of the existing equipment in the laboratory, the strength grade of the steel tube was chosen as Q355, and the dimensions were 100 mm × 100 mm × 4 mm. The strength grade of the steel hoop was Q235, the width was 80 mm, and the thickness was 8 mm. Self-compacting commercial concrete of strength class C40 was used for the filled concrete. The detailed design parameters for each specimen are shown in Table 1.

The manufacturing process of the specimen is as follows. First, the square steel tube was spot welded together and positioned in L- and T-shaped sections. Next, the steel hoop was welded to the steel tube wall of the hoop-type specimen by a fillet weld. The chamfers of the square steel tubes of the weld-type specimen were welded along the length direction, and welding seams connected the square steel tubes.

In addition, to ensure the effective transmission of axial loads, steel cover plates were set on the top and bottom ends of the column. After the special-shaped steel tube was made, the bottom cover plate was welded at one end, and concrete was poured into the steel tube. A vent hole was set at the bottom of each steel tube to ensure the quality of concrete pouring. After the concrete curing was completed, the upper-end concrete was ground and welded to the top cover plate. The manufacturing process of the L-shaped specimen is shown in Figure 3.

### 2.2. Material Properties

#### 2.2.1. Steel

The square steel tube used in the test was cold-formed, and the yield strength of steel in the corner area was higher than that in the flat area. In conformity with the requirements of the Chinese Standard GB/T 2975-2018, the steel in the flat area was selected for tensile samples. The sampling location and tensile samples are depicted in Figure 4.

In Table 2, the mechanical properties of steel with different thicknesses are provided, and the corresponding stress–strain curve of a 4-mm thick tensile sample example is shown in Figure 5.

#### 2.2.2. Concrete

The core concrete for the steel tubes is self-compacting commercial concrete, and the concrete mix proportions are as follows: cement: 295 kg/m^3^; fly ash 157 kg/m^3^; water: 172 kg/m^3^; sand 768 kg/m^3^; coarse aggregate: 756 kg/m^3^; and high range water reducer: 7.3 kg/m^3^. The Chinese standard GB/T 50081-2019 was followed while testing the mechanical properties of core concrete, and the test block is a cube with a side width of 150 mm. Concrete cube test pieces were poured simultaneously as the concrete in the steel tube and cured under the same conditions as the test specimen. The average compressive strength of the concrete cubes was 44.8 MPa.

### 2.3. Test Device and Measurement Scheme

The test was carried out using a 5000 kN universal hydraulic press. To imitate the hinge boundary, column hinge supports were placed at the top and bottom ends of the specimen. The loading method was displacement controlled. The speed of lifting the bottom plate of the hydraulic press was 0.5 mm/min. The controlled displacement was held for 2 min per stage, and the controlled displacement was 1/20 of the calculated limit displacement. The loading process stopped when the load reached less than 85% of the maximum bearing capacity. Simultaneously, if the column hinge articulation or the top plate of the test machine showed evident rotation, which poses a safety concern, then the loading operation was halted. The loading device schematic and a test photo are seen in Figure 6.

During the test, the load and longitudinal displacement data were obtained directly from the press. A linear variable displacement transducer was set in the middle of the column and at 1/4 of the height to measure the lateral deformation. Longitudinal and transverse strain gauges were applied to the tube’s surface to measure the strain. The layouts of the measuring device for the L- and T-shaped CFSST composite column with steel hoops are shown in Figure 7.

## 3. Test Results and Analyses

### 3.1. Test Phenomena and Failure Modes

Figure 8 depicts each surface’s numbering and view orientation to aid in describing the experimental phenomena. Meanwhile, the SS-CFST composite column does not have a clear boundary between slender and short columns, and it is defined as a stub column (*L/N* ≤ 3) and a slender column (*L/N* > 3). *L* means the height of the column and *N* refers to the maximum width of the cross-section.

The test phenomena of the hoop-type specimens (HL-1, HL-2, HT-1, and HT-2) and weld-type stub column specimens (WL-1 and WT-1) were similar. Here, the HL-1 specimen was used as an example to describe the test phenomena. The test piece showed no apparent changes from the beginning of loading to the load reaching 2822.6 kN. When the load reached 2946.2 kN, the steel tube wall in the middle of area II of faces F2, A1, and A2 showed slight bulging deformations, and the bulging degree continued to increase along with the loading. When the load reached 3133.8 kN, the steel tube in the middle of area I on faces A1, F1, and F2 underwent bulging deformation, and the tube in the middle of area II on faces B and C showed bulging deformation. When the load reached the peak carrying capacity of 3140.3 kN, it entered the downward phase, after which the outer steel tube wall was deformed with varying degrees of bulging at several locations. When the load was reduced to 2995.4 kN, the steel tube bulge deformation in the middle of area I on faces F1 and A1 formed a “bulge strip” at the corner. When the load was reduced to 2875.5 kN, the steel tube in the middle of area II of faces F1 and A1 also formed a “bulge strip” at the corner. Subsequently, the specimen was bent around the *y*-axis, the load was reduced to less than 85% of the ultimate load capacity, and the loading ended.

For the weld-type slender column, the WT-2 specimen was used as an example to describe the test phenomena. Changes in the specimen were not noticeable until the load increased to 2685 kN. The specimen rattled when the load was increased to 2864 kN. When the load increased to 3400 kN, a slight bending deformation occurred, and the magnitude of the bending deformation increased further with continued loading. After the load had risen to the ultimate load capacity of 3580 kN, it began to fall. The overall bending deformation became more pronounced and bulging occurred in the middle of the A2 surface, whereas the other surfaces remained unchanged. Subsequently, the overall bending developed rapidly, and the load was reduced to less than 85% of the peak carrying capacity, and the loading ended. The failure mode for each surface of the specimen is shown in Figure 9.

Thus, with the increase in compression deformation, the hoop-type specimens (HL-1, HL-2, HT-1, and HT-2) and weld-type stub specimens (WL-1 and WT-1) showed local buckling deformations on multiple outer steel tube walls during loading, as shown in Figure 10a. The scope of the local buckling of hoop-type specimens was broader, and individual samples even formed “bulge strips”, as shown in Figure 10b. Multiple bulging deformations were partly induced by the buckling of the steel tube and partly by the out-of-plane deformation of the crushed concrete. Part of the steel tube failed as the scope of the local bulge of the steel tube grew larger. Redistribution of cross-sectional stress occurred, from full cross-sectional compression to partial compression, partial tension, and overall bending of the specimen. The failure mode of the sample was dominated by local bulging and bending.

For the weld-type slender column specimens (WL-2 and WT-2), the local bulging of the steel tube was not apparent after reaching the peak carrying capacity, whereas the redistribution of the stress in the section occurred after yielding owing to the *P-δ* effect. The lateral deformation developed rapidly. The failure mode was mainly overall bending, and the greater the height of the specimen, the more pronounced the bending damage characteristics.

It should be noted that steel hoops did not break throughout the loading process and that the fillet welds connecting the steel hoops to the steel tubes and the connection welds between the square steel tubes did not crack. The square steel tubes did not separate from each other. The steel tube showed good synergistic working properties.

### 3.2. Carrying Capacity

Table 1 and Figure 11 show each specimen’s maximum carrying capacity under axial load. The peak loads of specimens HL-1, HT-1, and HL-2 increased by 19.86%, 20.86%, and 20.00%, respectively, compared with those of WL-1, WT-1, and WL-2. Therefore, the hoop-type member had a larger carrying capacity for the same cross-sectional dimensions and slenderness ratios than the weld-type specimen. The steel hoops’ setting improved the steel tube’s restraint effects on the concrete, delaying the local buckling of the steel tube. The bearing capacity of specimens HL-1 and WL-1 increased by 16.8% and 17.0%, respectively, compared with samples HL-2 and WL-2. The slenderness ratio of the specimen influenced the carrying capacity of the test piece. The smaller the slenderness ratio, the higher the bearing capacity. In addition, the carrying capacity of the specimen was directly proportional to the steel tube’s thickness, as shown by comparing the peak loads of WT-1 and WT-2. The greater the wall thickness of the tube, the higher the load capacity.

### 3.3. Load-Longitudinal Displacement Curves

Figure 12 presents the load (*F*)-longitudinal displacement (*Δ*) curve for each specimen. Before the steel yielded, the load-longitudinal displacement curve of the sample was basically a straight line, and the two changed linearly. After the steel tube entered the plastic stage, the axial compressive stiffness of the specimen reduced as the load increased, and the slope of the curve decreased. However, at this time, the steel tube was still in the strengthening stage, and the axial compressive load of the specimen was still increasing gradually. When the specimen reached the ultimate bearing capacity, the load decreased, and finally, the steel tube suffered from multi-face bulging failure or bending instability failure.

After reaching the peak load, the stub column specimen, especially the hoop-type stub member, still provided lateral restraint to the core concrete. The bearing capacity decreased slowly, and the slope of the *F-Δ* curve was small. When the compression deformation reached a certain level, the column hinge support rotated, and the specimen underwent bending deformation. The slender columns were more sensitive to initial defects (initial eccentricity and initial bending) and were more prone to bending instability under the influence of the *P-δ* effect. Furthermore, the load drop rate of the hoop-type slender column was faster than that of the weld-type specimen, and the slope of the *F-**Δ* curve was larger. This occurred because the multi-faceted local buckling deformation had already happened in the steel hoop-type slender specimen before the bending instability, and partial steel had been withdrawn from the work.

Additionally, the sequence of local buckling, peak load, and bending instability of the steel tube is shown in Figure 12. After the specimen entered the yielding phase, the surface of the steel tube appeared to be locally elevated, and the phenomenon of multi-surface local buckling occurred during the bearing capacity reduction phase. Therefore, the buckling of the steel tube represented nonlinear buckling, and the buckling stress was greater than the yield stress of the steel. Additionally, the ratio of the local buckling load to the peak carrying capacity of the hoop-type specimen was higher than that of the weld-type; consequently, the setting of the steel hoops delayed the local buckling of the steel tube.

### 3.4. Load–Strain Curves

Taking the HL-1 specimen as an example, the strain variations at different parts of the four sides of B, C, D, and F1 during the axial compressive loading process are shown in Figure 13. The positive strain value in the illustration indicates that the steel tube is under tension, while the negative value indicates that it is under compression. The longitudinal strain was used to observe the yield and buckling state of the steel tube, and the transverse strain was used to observe the circumferential stress of the steel and the restraint effect on concrete.

In the elastic phase, the *N*/*N*_u_-*ε* curve was basically an inclined straight line. The longitudinal strain was negative, and the transverse strain was positive, indicating a stress condition of longitudinal compression and transverse tension in the steel tube. The curve slopes of B-T-2 and B-B-2 were close to each other and significantly higher than those of B-M-2. The longitudinal stress at the hoops was smaller under the same load. The slope of the longitudinal strain curve was significantly higher than that of the transverse strain. The steel tube’s transverse strain was caused by longitudinal compression via the Poisson effect, and the steel tube’s transverse strain was minor.

In the elastic–plastic phase, the steel tube’s longitudinal strain at B-B first reached the yield strain, and then the transverse strain reached the yield strain. The steel entered a strengthening phase. At this point, the load was approximately 60% of the ultimate load. Thus, some tubes entered the strengthening phase before the specimen reached peak load. The increase in load capacity was not entirely due to the strengthening of the steel. The steel tubes provided sufficient lateral restraint for the core concrete during the strengthening phase, and the axial compression load continued to increase. When the load increased to 94% of the peak load, there was a significant bulge deformation of the steel tube wall. The bond between the steel tube and the concrete was broken. At this point, the transverse strain was caused by the local bulge of the steel tube and did not correctly reflect the restraint effects of the steel on the concrete. The strain of the steel tube at B-B-1 increased rapidly and reached the upper limit of the strain collector. The strain of B-B-2 changed from a negative to a positive value and quickly reached the upper limit of the strain collector. The stress state of the steel tube at position B-B changed from longitudinal compression and transverse tension to longitudinal and transverse tension in the plane stress state. After local buckling had occurred, the restraining effect of the steel tube on concrete was reduced, some of the concrete was crushed, and the specimen reached its ultimate carrying capacity.

In the failure stage, the steel tube on the F1 surface appeared to have a bulging deformation, and the strains of F1-T-2 and F1-B-2 changed from negative to positive. However, the steel tube at F1-T formed a band-shaped bulge, as shown in Figure 10b, whereas at F1-B, a bag-shaped bulge was formed, as shown in Figure 10a. The longitudinal deformation of the band-shaped drum was smaller than that of the bag-shaped drum, and the lateral deformation was more significant than that of the bag-shaped drum. The strain change of F1-T-1 was more extensive than that of F1-B-1, and the strain change of F1-T-2 was smaller than that of F1-B-2. Owing to the bending deformation of the specimen during the unloading stage, the strain of B-T-2 gradually changed from negative to positive, and the steel tube changed from longitudinal compression to longitudinal tension.

There were no obvious local bulge deformations in the C and D surfaces during the loading process, and their transverse strains were mainly caused by the Poisson effect, with a small range of variation. During the whole loading process, the longitudinal and transverse strains at each side of the steel hoops were small and did not reach the yield strain. Due to the bending deformation, the longitudinal strains at steel hoops C and D on the tensile side changed from negative to positive. In contrast, the transverse strains changed from positive to negative during the damage phase. The stress state of the steel hoops changed from longitudinal tension and transverse compression to longitudinal compression and transverse tension.

### 3.5. Poisson’s Ratio for Steel Tubes

The transverse and longitudinal strain ratios of steel tubes at different positions of some specimens during loading are shown in Figure 14. The initial Poisson of steel was greater than that of concrete at the beginning of loading. The transverse deformation of the steel tube was greater than that of concrete after the same longitudinal compression deformation. At this time, the steel tube had no lateral restraint effects on the concrete. With the progress of loading, the internal micro-cracks of concrete gradually increased, the volume expanded laterally, the Poisson’s ratio also steadily increased, and the lateral restraint effects of steel tube on concrete appeared and increased gradually.

From the beginning of loading until approximately 90% of the peak carrying capacity, the transverse and longitudinal strain ratio at A2-T of the HL-1 specimen did not change greatly, and the value remained at approximately 0.3, which was close to the initial Poisson’s ratio of the steel. After the load reached 0.9 times the ultimate load, the steel tube buckled locally at the A2-T position, and the transverse and longitudinal strain ratio changed abruptly. With the increase in load, the transverse and longitudinal strain ratios of the steel tube at the B-T position of the HL-1 specimen, A2-M position of the WL-1 specimen, and D-B position of the HL-2 specimen increased along with the load ratio, indicating that the restraint effects of the steel tube on concrete gradually strengthened. In particular, when approaching the ultimate bearing capacity, the steel tube’s strain ratio increased rapidly due to the local buckling of some steel tubes and the crushing of concrete.

### 3.6. Bearing Mechanism

At the start of the loading process, the steel tube and concrete mainly assumed the longitudinal compressive stress. At that time, the two worked independently, and the combined action had not yet occurred. The bearing capacity of the specimen can be regarded as the superposition of the two, and the specimen will be in a state of elasticity.

With continuous loading, the micro-cracks in the concrete continued to develop, and the Poisson’s ratio continued to increase. The concrete was in the stress state of longitudinal compression, circumferential and radial tension. When the transverse deformation of the concrete exceeded the transverse deformation of the steel tube, the steel tube restricted the concrete, and the concrete changed to triaxial compression. A plane stress state of circumferential tension and longitudinal compression could be applied to the steel tube. The steel tube surface of the stub column specimen appeared to swell locally as the loading progressed, the growth rate of the load slowed down, and the axial compression stiffness decreased. The specimen was in an elastic–plastic state. For the slender column, the specimen was susceptible to lateral deformation after entering the elastic–plastic phase, whereas the *P-δ* second-order effect caused the additional bending moment to increase rapidly. When the additional bending moment was the same as the resisting moment of the section, the load reached the maximum value. During the damage phase, the steel tube’s local buckling of the stub column was further extended, and some of the tubes withdrew from the work. However, some of the steel tubes still provided restraint to the core concrete at this time, resulting in a slight decrease in the load-carrying capacity. However, the slender column suffered bending instability damage under the action of the additional bending moment, resulting in a rapid decrease in the bearing capacity. Figure 15 shows the process of stress changes in steel tubes and concrete.

Core concrete can be divided into effectively confined and ineffectively confined areas due to stirrups’ restraint effects on core concrete. The steel tube’s restraining effects on core concrete are similar to a stirrup’s. The bending stiffness of the steel tube in the flat area was small, the bulging deformation occurred first in the flat area under the axial compression load, and the restraint effect on the core concrete was weak, as shown in Figure 16. However, the bending stiffness in the corner was large, and the deformation was small. Under the action of mutually perpendicular tension, strong constraints were formed on the core concrete along the diagonal direction [15,16].

At the same time, the above test phenomena revealed that the local bulge deformation of the specimen occurred mainly on the outside of the shaped steel tube. In contrast, no local buckling phenomenon occurred in the connective coincidence area between the steel tubes and the position of the hoops. Therefore, the steel tube connection coincidence area and hoop positions can be regarded as a strong constraint boundary, whereas the internal concrete can be regarded as an effectively confined area. The confining mechanism of the steel tube to the core concrete in the L-shaped CFSST composite column with steel hoops is shown in Figure 17.

## 4. Finite Element Models

### 4.1. Stress–Strain Relationships for Steel

In this research, the finite element software ABAQUS was selected to evaluate the mechanical properties of SS-CFSST composite columns under axial compression. The von Mises yield criterion was used to define the constitutive model of steel, which was then linked to a flow rule with isotropic strain hardening. The stress–strain relationship of steel was as follows: (1)$$\sigma_{i}=\{\begin{array}{cccc} E_{s}\varepsilon_{i} & & \varepsilon_{i}\leq \varepsilon_{y} & \\ f_{y} & & \varepsilon_{y}<\varepsilon_{i}\leq \varepsilon_{\mathrm{st}} & \\ f_{y}+\zeta E_{s}(\varepsilon_{i}-\varepsilon_{\mathrm{st}}) & & \varepsilon_{\mathrm{st}}<\varepsilon_{i}\leq \varepsilon_{u} & \\ f_{u} & & \varepsilon_{i}>\varepsilon_{u} & \end{array},$$
where *σ*_i_ and *ε*_i_ represent the equivalent stress and strain, respectively; *f*_y_ and *f*_u_ indicate the yield strength and ultimate strength, respectively; *ε*_y_ and *ε*_u_ represent the strain when the stress achieves the yield strength and ultimate strength, respectively; *ε*_st_ represents the strain when the steel was strengthened, where *ε*_st_ = 12*ε*_y_ and *ε*_u_ = 120*ε*_y_; and *E*_s_ and *E*_st_ represent the elastic modulus and strengthening modulus of steel, respectively, with *E*_s_ = *ζE*_st_ and ζ = 1/216.

### 4.2. Element Type, Interaction, and Boundary Conditions

In the finite element model, the SS-CFSST composite column with steel hoops consisted of four main parts: steel tube, steel hoop, core concrete, and cover plate. To approximate the steel tube and steel hoop, a four-node shell element was used, each node had a simplified integral and six degrees of freedom (S4R). Concrete has a simplified integral solid element with eight nodes, each with three translational degrees of freedom (C3D8R) [17,18]. Due to the large thickness, a rigid body element was used to simulate the cover plate.

Surface-to-surface contact exists between the steel tube and the core concrete. The penalty function simulated the tangential interaction, and the friction coefficient was 0.6 [19]. The normal interaction is simulated by “hard contact”. The interaction between the cover plate and concrete and the steel tube and steel hoop was also surface-to-surface contact. The difference in the contact between steel tube and concrete was that the coefficient of friction was defined as 0.3. In addition, the weld seams were modeled with a “tie” constraint. Additionally, small slip assumptions were used, and the shell’s thickness was neglected. For the steel and core concrete, the Poisson’s fractions were 0.3 and 0.2, respectively [20,21].

Depending on the column hinge support position in the test, a loading line was defined on the cover plate, and a reference point (RP) was set outside the loading line. The RP was coupled to the loading line. The vertical displacement along the *Z*-axis was applied to the loading line through the RP. The displacements along the X and Y axes of the top cover RP and its rotation around the *Z*-axis were limited, as were the bottom cover RP’s displacements along the X, Y, and Z axes and its rotation around the *Z*-axis. The loading process in finite element analysis (FEA) should be consistent with the test loading conditions as far as possible. The boundary condition of the SS-CFSST composite column is shown in Figure 18.

### 4.3. Stress–Strain Relationships for Confined Concrete

The confinement mechanism of the steel tube on the core concrete should be appropriately evaluated to determine the constitutive relation of confined concrete. Susantha [22], Sheikh [23], and Mander [24] established the analytical models for the confinement mechanisms by dividing the section into an effectively confined area and an ineffectively confined area in tied columns and proposed the stress–strain relationships for concrete confined by rectangular hoop reinforcement. Han [25] proposed the coefficient of confinement effect *ξ* to evaluate the confinement strength of steel pipe on concrete and established the stress–strain relationship of CFST by fitting a large amount of experimental data. Ding [26] analyzed the elastic–plastic process of CFST columns using the continuum media mechanics approach and established the stress–strain relationship for concrete under axisymmetric triaxial compression.

Based on the confining mechanism of the SS-CFSST composite column established above, the core concrete’s force characteristics were similar to those of rectangular stirrup-confined concrete. Therefore, The SS-CFSST column confined concrete stress–strain model adopted in this paper is based on Mander’s stirrup-confined concrete constitutive model, whereas, considering the characteristics of the special-shaped steel tube on the core concrete. The stress–strain relationship of the core concrete was as follows:(2)$$\sigma =\frac{f_{\mathrm{cc}}\cdot x\cdot r}{r-1+x^{r}}$$
(3)$$x=\frac{\varepsilon }{\varepsilon_{\mathrm{cc}}}$$
(4)$$r=\frac{E_{c}}{\left(E_{c}-E_{\sec}\right)}$$
(5)$$\varepsilon_{\mathrm{cc}}=\varepsilon_{c0}\cdot \left[1+5\left(\frac{f_{\mathrm{cc}}}{f_{c0}}-1\right)\right]$$
(6)$$f_{\mathrm{cc}}=f_{c0}\cdot \left(-1.254+2.254\sqrt{1+\frac{7.94f_{1}}{f_{c0}}}-2\frac{f_{1}}{f_{c0}}\right)$$
(7)$$f_{1}=K_{e}\cdot f_{1}^{\prime }=\frac{K_{e}\cdot f_{\mathrm{sh}}\cdot 2t_{s}}{\left(a-2t_{s}\right)}$$
where *σ* and *ε* represent the stress and strain, respectively, of the confined concrete; *f*_cc_ and *ε*_cc_ represent the ultimate stress and the corresponding strain of the confined concrete, respectively; *r* represents the curve-shape parameter; *f*_c0_ and ε_c0_ represent the maximum stress and the corresponding strain, respectively, of the unconstrained concrete, with ε_c0_ = 0.002; *f*_1_ represents the effective lateral compressive stress; *E*_c_ represents the initial elastic modulus of concrete; *E*_sec_ represents the secant modulus at the maximum stress of confined concrete; *K*_e_ represents the effective lateral restraint coefficient, which was used to reflect the restraint effect of steel tubes on core concrete. The calculation formula of *K*_e_ was as follows:(8)$$K_{e}=k_{e1}\times k_{e2}$$
(9)$$k_{e1}=\frac{A_{q}}{A_{1}}$$
where *k*_e1_ represents the cross-sectional restraint factor, *k*_e2_ represents the lateral restraint factor, *A*_q_ represents the area of concrete in the effectively confined area, and *A*_1_ represents the total area of concrete. In order to simplify the calculation, the following assumptions were established when calculating the *K*_e_ of the SS-CFSST composite column with steel hoops:CFST columns can be regarded as stirrups with zero longitudinal spacing between stirrups; consequently, its lateral effective restraint coefficient *k*_e2_ was taken as 1;The effective lateral restraint coefficient was calculated using the ratio of the concrete volume in the effectively confined area to the total concrete volume;The longitudinal deformation of the core and limb columns was coordinated under axial compression loading;The dividing line between the effectively and ineffectively confined zones of concrete in the cross-section of the square steel tube was assumed to be a parabola with a starting angle *θ* of 45° [23].

Based on the above assumptions, the concrete volume in the ineffectively confined area in the SS-CFSST can be simplified as the product of the area and height of the ineffectively confined area of the cross-section. Thus, the effective lateral restraint effect coefficient for the limb column (*K*_el_) was as follows:(10)$$k_{\mathrm{el}}=\frac{V_{\mathrm{ql}}}{V_{1}}=1-\frac{\sum_{i=1}^{n}(a-2t_{s}-2r{)}^{2}\cdot D_{i}}{2\cdot L\cdot \left[(a-2t_{s}-2r{)}^{2}+4(a-2t_{s}-2r)r+\pi r^{2}\right]}$$

The effective lateral restraint coefficients for L-shaped and T-shaped section core columns were as follows:(11)$$k_{e-\mathrm{Lc}}=\frac{V_{q-\mathrm{Lc}}}{V_{1}}=1-\frac{\sum_{i=1}^{n}(a-2t_{s}-2r{)}^{2}\cdot D_{i}}{3\cdot L\cdot \left[(a-2t_{s}-2r{)}^{2}+4(a-2t_{s}-2r)r+\pi r^{2}\right]}$$
(12)$$k_{e-\mathrm{Tc}}=\frac{V_{q-\mathrm{Tc}}}{V_{1}}=1-\frac{\sum_{i=1}^{n}(a-2t_{s}-2r{)}^{2}\cdot D_{i}}{6\cdot L\cdot \left[(a-2t_{s}-2r{)}^{2}+4(a-2t_{s}-2r)r+\pi r^{2}\right]}$$
where *D* represents the net distance between adjacent hoops, *L* represents the height of the column, for weld-type SS-CFSST composite column, $\sum_{i=1}^{n}D_{i}$ = *L*; *a* and *t*_s_ represent the steel tube’s side length and thickness, respectively; *r* is the corner radius.

## 5. Verification

### 5.1. Failure Mode Comparison

A comparison of failure modes between the experiment and FEAs of some specimens is shown in Figure 19. The failure mode of the HL-1 specimen simulated by the FEA was mainly a local bulge on the surface of the steel tube between the two steel hoops, with the specimen showing some degree of bending deformation. The failure pattern of the HT-1 specimen was a local bulge on the steel tube between the two steel hoops and on both sides of the hoops, but the bending deformation of the column was not noticeable. The damage mode of specimen WL-2 was mainly the bending deformation, and there was no evident local bulging on the surface of the steel tube. A comparison of the test damage morphology of each specimen revealed that the damage morphology of the finite element simulation was similar to that of the test.

### 5.2. Load–Strain Curves Comparison

Figure 20 compares the load-mean longitudinal strain curves of some specimens obtained from numerical analysis with the test curves. The two curves basically coincide in the elastic phase. The development trend of the two curves also remains the same in the elastic–plastic and damage stages. Consequently, the average longitudinal and transverse strains in the column obtained from the finite element simulation were in good agreement with the actual test results.

### 5.3. Carrying Capacity Comparison

The comparison of the carrying capacity of each specimen calculated using different constitutive relations and the measured value of the test are shown in Table 3 and Figure 21. *N*_p_,_fea_ is the peak carrying capacity obtained from FEAs using the stress–strain relationships of confined concrete are presented in this paper, and references [25,26], respectively. The average values of the carrying capacity of the specimen calculated by the above three types of confined concrete’s constitutive relations and the test values of the test were 1.01, 0.96, and 0.89, respectively, and the standard deviation was 0.069, 0.088, and 0.099, respectively. The peak carrying capacity of each specimen calculated by the constitutive relation in this research was the closest to the actual test results owing to considering the restraining effects of the steel hoop and the steel tube connection coincidence area on the core concrete. In contrast, only the square steel tube’s constraint effect was considered in the reference [25,26].

On the basis of the comparisons of the above failure modes, load–strain curves, and bearing capacity, using the finite element software ABAQUS and selecting a reasonable constitutive relationship between steel and concrete, the axial compression behavior of the SS-CFSST composite column can be simulated more accurately using our model compared with others.

## 6. Conclusions

In this paper, axial compression tests of eight SS-CFSST column specimens were conducted to investigate the mechanical properties of the specimens and analyze the bearing mechanism. On this basis, the corresponding finite element model was established, and the accuracy of the finite element model was verified. The main conclusions are as follows:The strength damage was dominated by the hoop-type specimen and the weld-type stub column. The damage mode of weld-type slender columns was dominated by bending instability damage.The carrying capacity of specimens HL-1 and WL-1 increased by 16.8% and 17.0%, respectively, compared with HL-2 and WL-2. However, the peak loads of specimens WT-1 decreased by 12.29% compared with WT-2. Thus, the bearing capacities of SS-CFSST columns were directly proportional to the steel tube’s thickness and inversely proportional to the slenderness ratios of the specimen.The peak loads of specimens HL-1, HT-1, and HL-2 increased by 19.86%, 20.86%, and 20.00% compared with WL-1, WT-1, and WL-2, respectively. The steel hoops can not only be regarded as a connection of square steel tubes but also delays the buckling of the steel tubes and improve the carrying capacity of the specimen.The average value of the ratio between the carrying capacity obtained from the FEM using the proposed constitutive relationship of confined concrete and the tested values is 1.01, with a standard deviation of 0.069. The finite element model established in this paper can be used to simulate the mechanical properties of SS-CFSST columns under axial compression.

## Figures and Tables

**Figure 1 materials-15-04179-f001:**
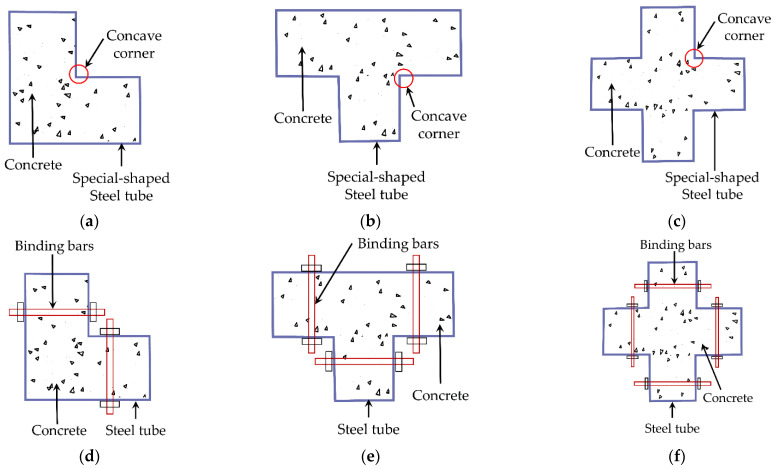
SS-CFSST composite columns (**a**) traditional L-shaped; (**b**) traditional T-shaped; (**c**) traditional cross-shaped; (**d**) L-shaped with binding bars; (**e**) T-shaped with binding bars; (**f**) cross-shaped with binding bars; (**g**) L-shaped with stiffeners; (**h**) T-shaped with stiffeners; (**i**) cross-shaped with stiffeners.

**Figure 2 materials-15-04179-f002:**
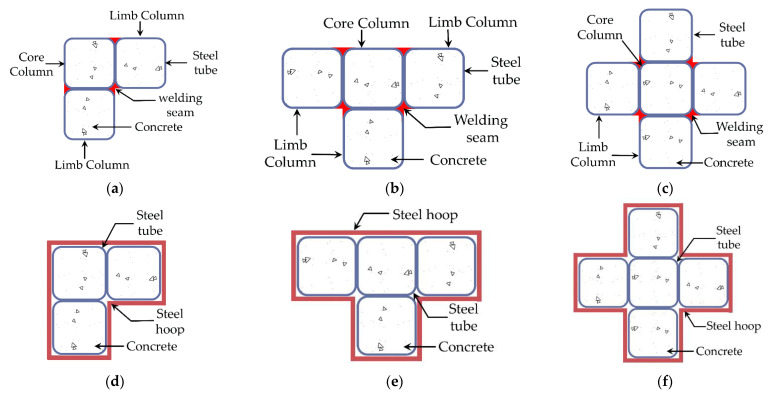
Section diagram of the SS-CFSST composite columns (**a**) weld-type L-shaped; (**b**) weld-type T-shaped; (**c**) weld-type cross-shaped; (**d**) hoop-type L-shaped; (**e**) hoop-type T-shaped; (**f**) hoop-type cross-shaped.

**Figure 3 materials-15-04179-f003:**
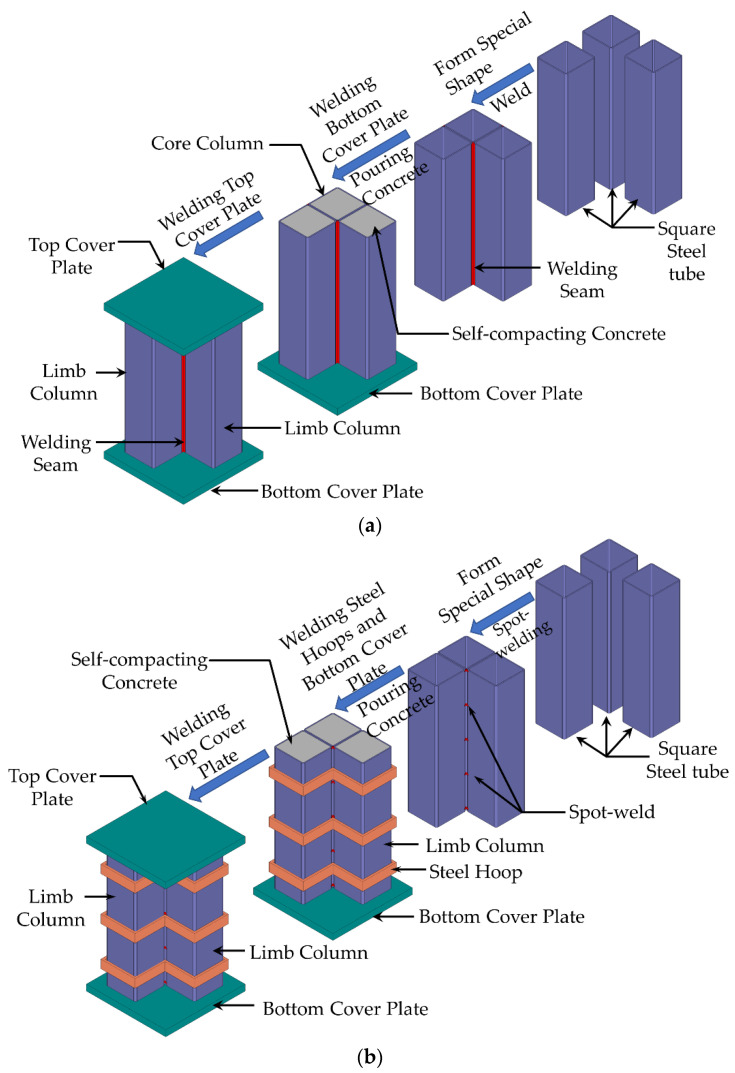
Schematic diagrams of the specimens manufacturing process with L-shaped sections (**a**) weld-type; (**b**) hoop-type.

**Figure 4 materials-15-04179-f004:**
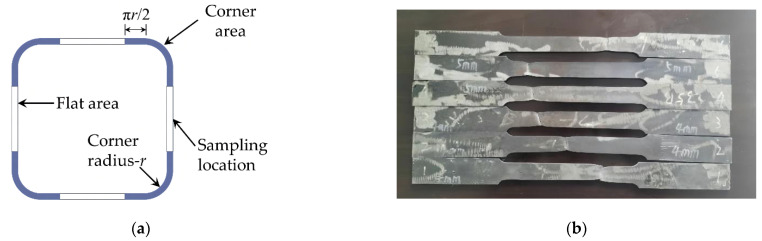
Tensile test (**a**) sampling location; (**b**) samples.

**Figure 5 materials-15-04179-f005:**
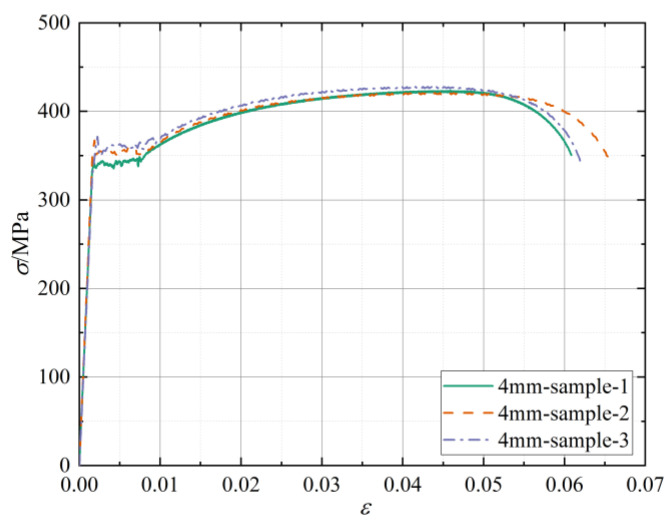
Stress–strain curve of 4-mm thick tensile steel sample.

**Figure 6 materials-15-04179-f006:**
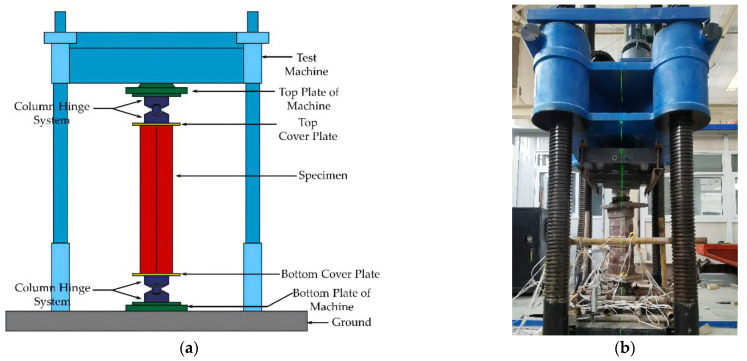
Test device (**a**) schematic diagram of loading device; (**b**) test photo.

**Figure 7 materials-15-04179-f007:**
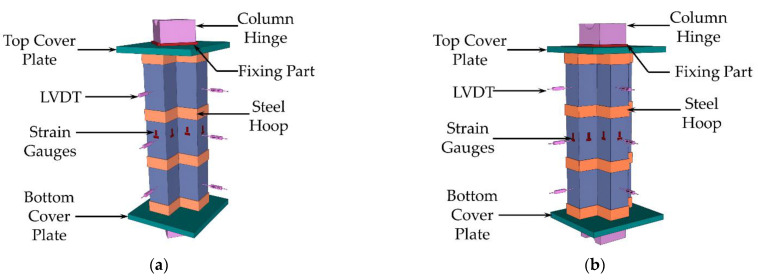
Layouts of the measuring device (**a**) L-shaped; (**b**) T-shaped.

**Figure 8 materials-15-04179-f008:**
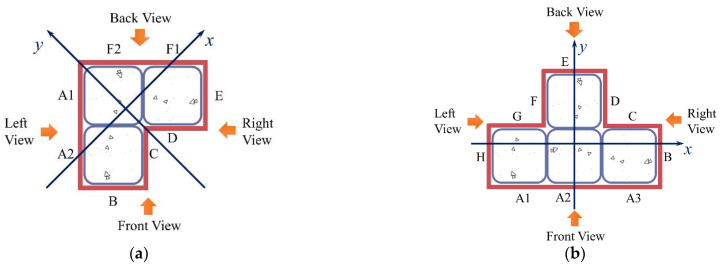
Numbering and view orientation of each surface. (**a**) L-shaped; (**b**) T-shaped.

**Figure 9 materials-15-04179-f009:**
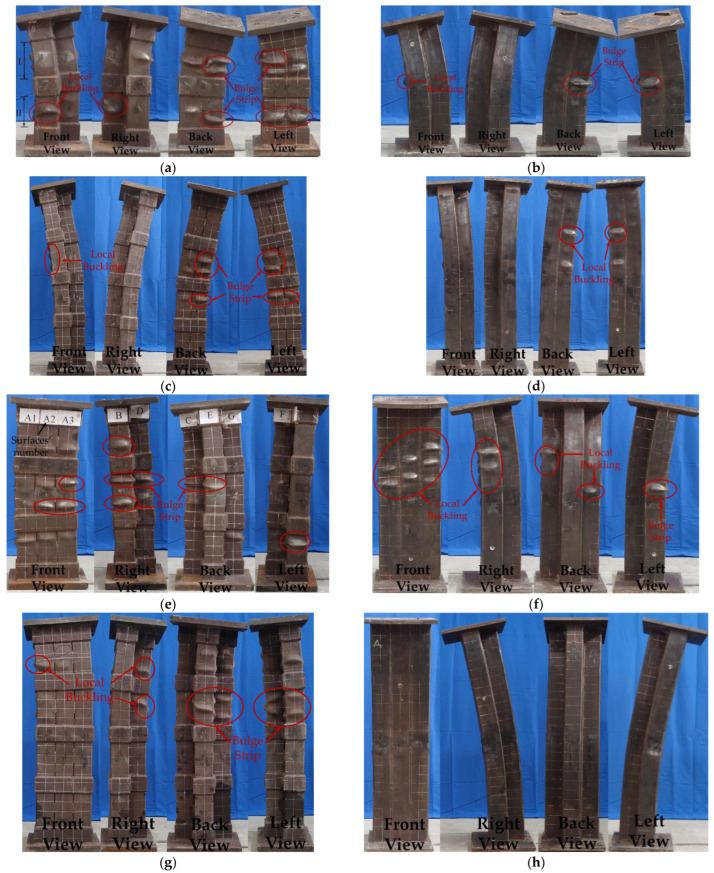
Failure modes of test specimens (**a**) HL-1; (**b**) WL-1; (**c**) HL-2; (**d**) WL-2; (**e**) HT-1; (**f**) WT-1; (**g**) HT-2; (**h**) WT-2.

**Figure 10 materials-15-04179-f010:**
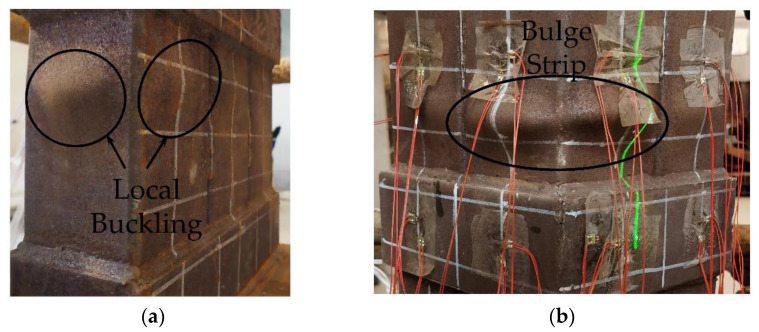
Bulging deformation (**a**) local buckling of HT-1; (**b**) bulge strip of HL-2.

**Figure 11 materials-15-04179-f011:**
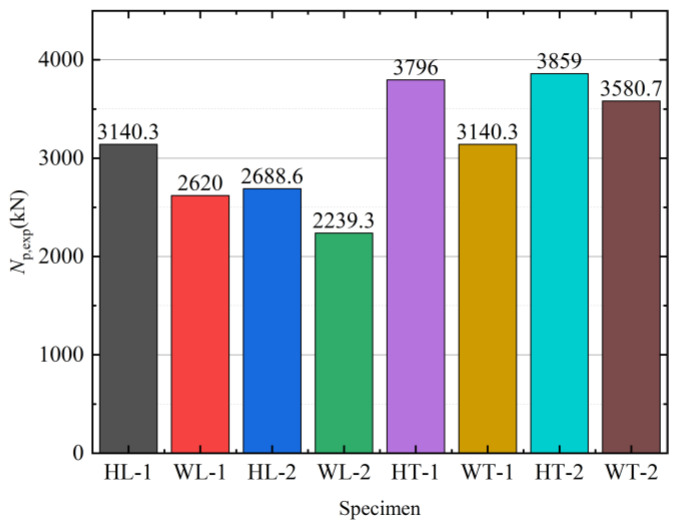
The peak load of test specimens.

**Figure 12 materials-15-04179-f012:**
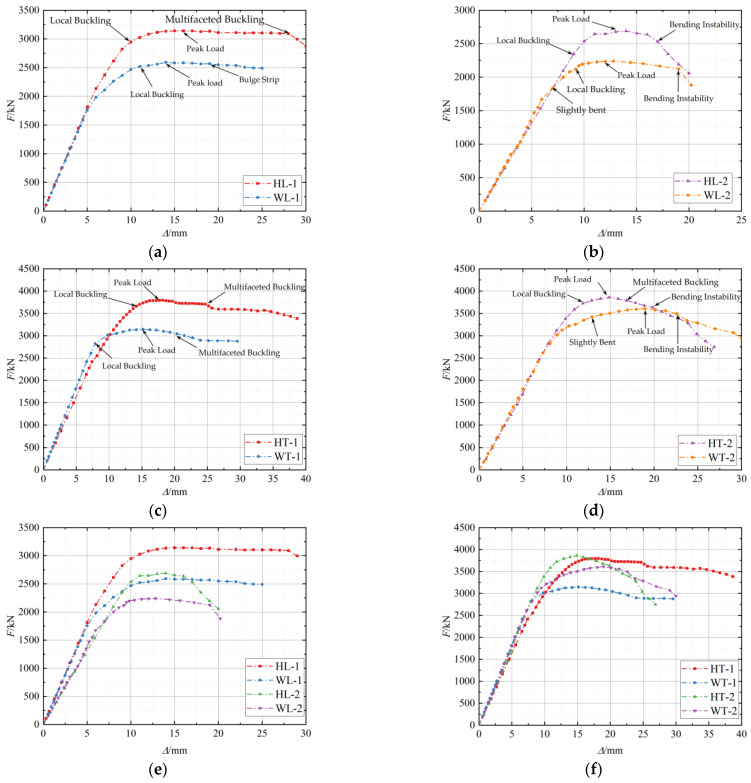
Load-longitudinal displacement curves (**a**) HL-1 and WL-1; (**b**) HL-2 and WL-2; (**c**) HT-1 and WT-1; (**d**) HT-2 and WT-2; (**e**) L-shaped; (**f**) T-shaped.

**Figure 13 materials-15-04179-f013:**
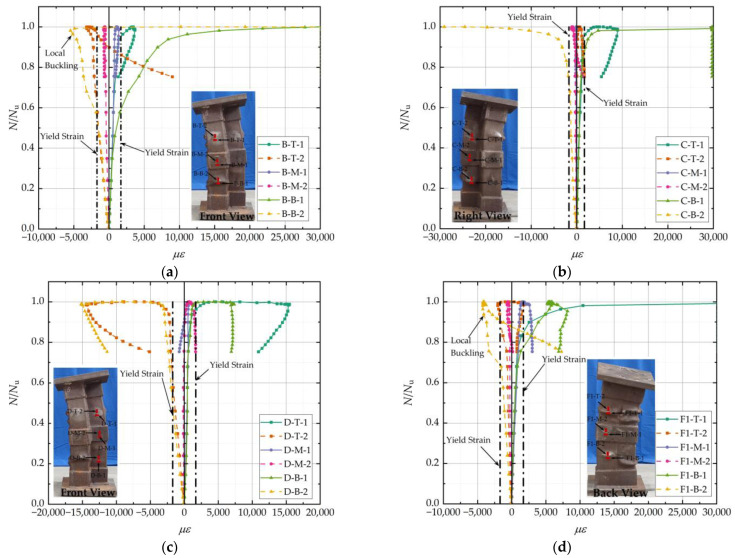
Load–strain curves of the HL-1 specimen (**a**) Surface B; (**b**) Surface C; (**c**) Surface D; (**d**) Surface F1.

**Figure 14 materials-15-04179-f014:**
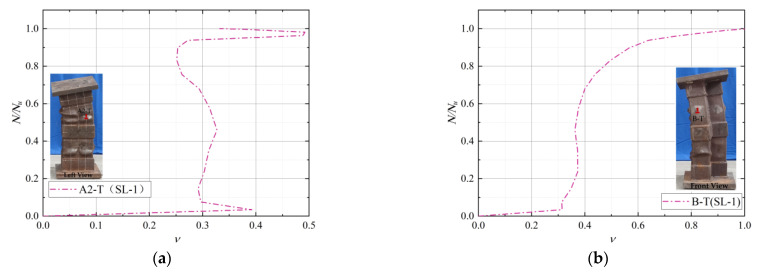
Poisson’s ratio of the steel tube (**a**) A2-T of HL-1; (**b**) B-T of HL-1; (**c**) A2-T of WL-1; (**d**) D-B of HL-2.

**Figure 15 materials-15-04179-f015:**
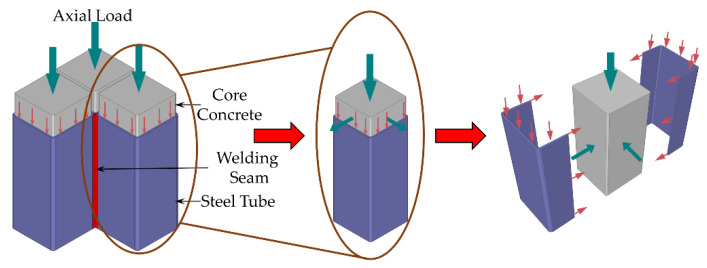
The process of stress changes in steel tubes and concrete.

**Figure 16 materials-15-04179-f016:**
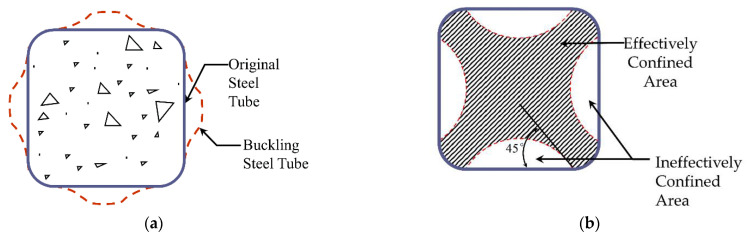
(**a**) Buckling characteristics of square steel tube under axial load; (**b**) effectively confined concrete in a square column.

**Figure 17 materials-15-04179-f017:**
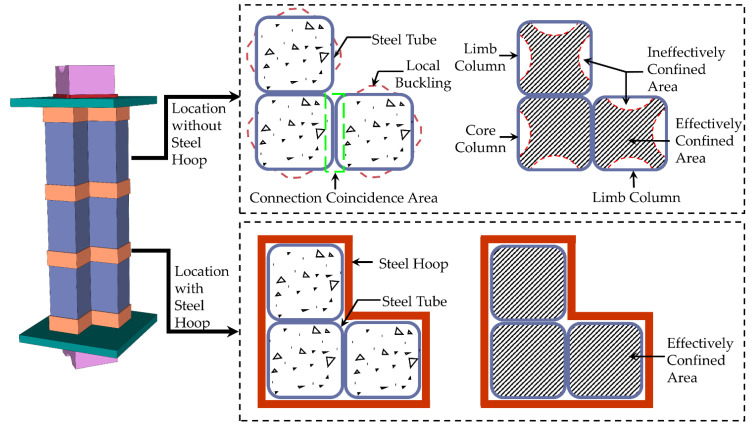
Confining mechanism of the steel tube to concrete in an L-shaped CFSST composite column with steel hoops.

**Figure 18 materials-15-04179-f018:**
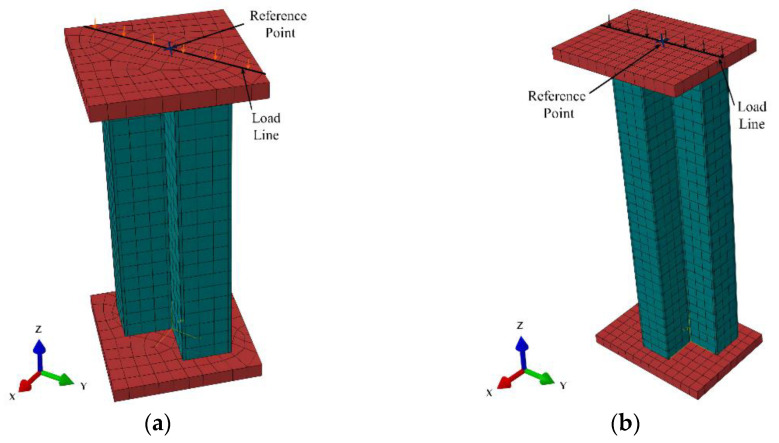
Boundary Conditions (**a**) L-shaped; (**b**) T-shaped.

**Figure 19 materials-15-04179-f019:**
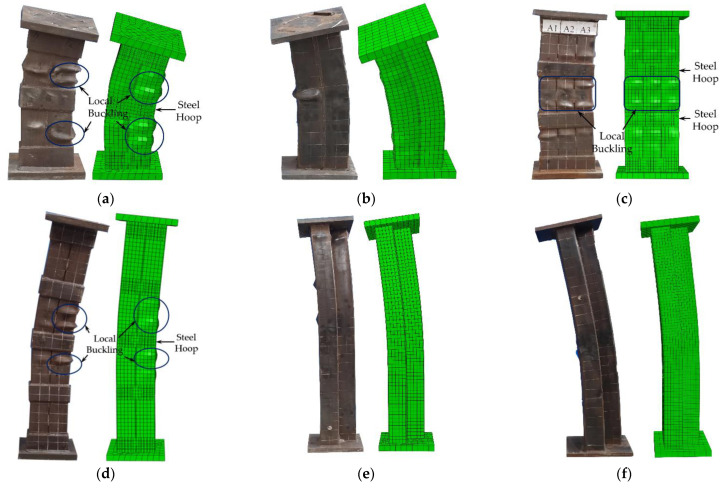
Comparison of failure modes between the test and FEA (**a**) HL-1; (**b**) WL-1; (**c**) HT-1; (**d**) HL-2; (**e**) WL-2; (**f**) WT-2.

**Figure 20 materials-15-04179-f020:**
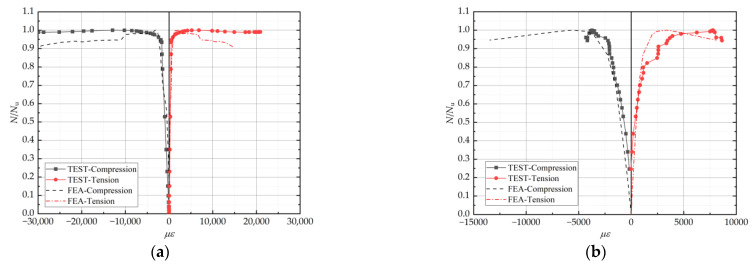
Comparison of load–strain curves between the test and FEA (**a**)HL-1; (**b**) WT-1.

**Figure 21 materials-15-04179-f021:**
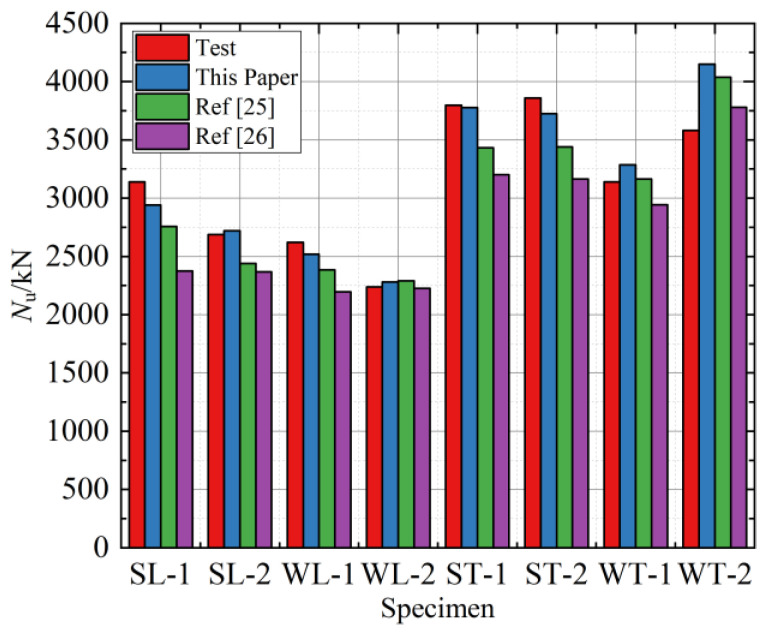
Ultimate bearing capacity comparison between test results and FEAs.

**Table 1 materials-15-04179-t001:** Main design parameters.

Specimens	*t*_s_ (mm)	*H* (mm)	*D* (mm)	*L* (mm)	*λ*	Steel Grade	Concrete Grade	*N* _p,exp_	Type
HL-1	4	80	260	600	14	Q355	C40	3140.3	Hoop
HL-2	4	80	280	1200	27	Q355	C40	2688.6	Hoop
WL-1	4	-	-	600	14	Q355	C40	2620.0	Weld
WL-2	4	-	-	1200	27	Q355	C40	2239.3	Weld
HT-1	4	80	260	900	16	Q355	C40	3796.0	Hoop
HT-2	4	80	280	1200	21	Q355	C40	3859.0	Hoop
WT-1	4	-	-	900	16	Q355	C40	3140.7	Weld
WT-2	5	-	-	1200	21	Q355	C40	3580.7	Weld

Note: *t*_s_ represents the thickness of the steel tube, *H* represents the width of the steel hoop, *D* represents the net distance between adjacent hoops, *L* represents the height of the column, *λ* represents the slenderness ratio of the column, *λ* = *L*/*i*_sc_, *i*_sc_ represents the radius of gyration, *i*_sc_ = (*I*_sc_/*A*_sc_)^1/2^, *I*_sc_ represents the bending moment of inertia, *A*_sc_ represents the cross-sectional area. *N*_p,exp_ means the peak load obtained by the test.

**Table 2 materials-15-04179-t002:** Mechanical properties of steel.

Sample	*t* (mm)	*B* (mm)	Steel Grade	*f*_y_ (MPa)	*ε*_y_ (με)	*f*_u_ (MPa)	*E* (MPa)	*u*
LS4	4.0	20	Q355	344	1699	423	202,560	0.82
LS5	5.0	20	Q355	390	1857	488	210,055	0.81
LS8	7.9	20	Q235	269	1353	435	199,011	0.62

Note: the number in the sample name represents the nominal thickness, and *t* represents the actual thickness. *B* denotes the width of the tensile sample, *f*_y_ and *ε*_y_ are the yield strength and the corresponding strain, respectively, *f*_u_ indicates the ultimate strength, *E* represents the elastic modulus, *u* represents the yield-strength ratio, and *u* = *f*_y_/*f*_u_.

**Table 3 materials-15-04179-t003:** Peak carrying capacity comparison between test results and FEA results.

Specimen	*N*_p,exp_ (kN)	*N*_p,fea_/*N*_p,exp_		
		This Paper	Reference [25]	Reference [26]
HL-1	3140.30	0.94	0.88	0.76
HL-2	2688.60	1.01	0.91	0.88
WL-1	2620.00	0.96	0.91	0.84
WL-2	2239.30	1.02	1.02	0.99
HT-1	3796.00	0.99	0.90	0.84
HT-2	3859.00	0.97	0.89	0.82
WT-1	3140.30	1.05	1.01	0.94
WT-2	3580.70	1.16	1.13	1.06
Average		1.01	0.96	0.89
Standard deviation		0.069	0.088	0.099

## Data Availability

Not applicable.

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
