# Peer review of "Experimental Investigation of Special-Shaped Concrete-Filled Square Steel Tube Composite Columns with Steel Hoops under Axial Loads"

_materials, 2022, doi:10.3390/ma15124179_

Round 1

Reviewer 1 Report

After carefully reviewing the article, my conclusion is that this research requires major modifications to achieve the publication quality standard of “Materials”. With the intention of helping the authors to improve their article, the following amendments are suggested:

The methodological approach compares the performance of two types of SS-CFST Columns under the action of compression loads. On one hand, there were specimens formed by 4 mm (and 5 mm) steel tubes welded together. On the other hand, there were specimens formed by 4 mm steel tubes attached together by 8mm thick and 80mm wide steel hoops. In my opinion this comparison is only partially valid, because the observed damage (bulge in the spaces between hoops) is totally predictable, and always observed in the 4mm (and 5mm) plates. A better approach would be compare the “hoop columns” with “no-hoop columns” which thickness would generate a steel tube volume equivalent to the “hoop columns”. For example, SL1 has a steel volume equals to 441.600 mm3 which is considerable higher than WL1 that has a steel volume equals to 288000 mm3. To obtain a “no hoop column” with equals steel volume than SL1, a thickness of 6mm should be considered. This way, the study compares SS-CFST elements with relative equivalent cost.

This does not necessarily means new tests. Instead, it is suggested to analyze this new case using ABAQUS models. These models have been validated in the paper and the results of this new analysis should be sufficiently robust to complete the study.

Of course, it is expected that these new results will help to improve the conclusions, which still sounds weak in its current state.

Author Response

Dear reviewer

Many thanks for your time in reading our manuscript and for giving us your insightful suggestions. All these are of great importance for improving the quality of this manuscript. We do value these suggestions and appreciate your constructive comments on our manuscript entitled “Experimental Investigation of Special-Shaped Concrete-Filled Square Steel Tube Composite Columns with Steel Hoops Under Axial Loads” (No. 1735322)

We have responded to your comments in the document. At the same time, we’ve made modifications to the manuscript in response to other reviewers’ suggestions and comments, which are highlighted in yellow in the corrected version. We hope that all these changes fulfill the requirements to make the revised manuscript acceptable for publication in Materials.

Thank you very much for your comments and suggestions.

Best regard

All Authors

Reviewer 2 Report

This paper presents a consideration of the Experimental Investigation of Special-Shaped Concrete-Filled
Square Steel Tube Composite Columns with Steel Hoops Under Axial Loads.
The work is interesting from the point of view of engineering and applications of numerical simulations in the evaluation of typical engineering problems. In my opinion, the work does not contain major flaws that could affect the positive evaluation of the research paper presented by the authors. Only a few issues would have to be improved for the content to be published:

  1. In the 3rd sentence of the introduction, 8 research papers [3-10] were referenced at once. Please expand the description so that one sentence does not cite such a large number of publications, but expand the description of what the authors of each paper contributed to the issues under consideration.
  2. The introduction refers to the phenomenon of structural buckling and the finite element method. It would be advisable to refer to research papers extensively presenting this type of subject matter: 10.1016/j.compositesb.2020.107931, 10.1002/nme.6757, 10.1016/j.compositesb.2013.10.080.
  3. Please clearly demonstrate at the end of the introduction the novelty of the presented work relative to many other thematically similar research papers.
  4. Figure 3a should be clearer, please take care of this for better readability. The same applies to Figure 4.
  5. Figure 16 is not very readable. The analysis was done in ABAQUS, so there are better ways to present the visualization of the model. Please take care of better presentation of this figure.
  6. The conclusions do not completely refer to the research results obtained. Conclusions were made partly based on main reflections and qualitative assessment. There is nothing about the quantitative evaluation in relation to numerical values, etc. Conclusions should clearly present the directly obtained research results on the basis of which a quantitative evaluation should be made.

Author Response

Dear reviewer

Many thanks for your time in reading our manuscript and for giving us your insightful suggestions. All these are of great importance for improving the quality of this manuscript. We do value these suggestions and appreciate your constructive comments on our manuscript entitled “Experimental Investigation of Special-Shaped Concrete-Filled Square Steel Tube Composite Columns with Steel Hoops Under Axial Loads” (No. 1735322)

We have responded to your comments one by one in the document. At the same time, we’ve made modifications to the manuscript in response to other reviewers’ suggestions and comments, which are highlighted in yellow in the corrected version. We hope that all these changes fulfill the requirements to make the revised manuscript acceptable for publication in Materials.

Thank you very much for your comments and suggestions.

Best regard

All Authors

Author Response

(The authors gave the same response as above.)

Reviewer 4 Report

  The present research investigates proposing a new type of special-shaped concrete-filled square steel tube (SS-CFSST) composite column, and eight specimens were tested under axial loads with section shape, construction method, slenderness ratio, steel-tube thickness and strength as variation parameters. The failure modes, peak load, load-displacement curves, load-strain curves and Poisson's ratio of the steel tubes were analyzed. The failure modes of hoop-type specimens and weld-type stub columns were mainly the local buckling of steel tubes and bending failure, and those of the weld-type slender columns were mainly overall bending failure.

The present work represents excellent performed work in the field of Materials Science & civil Engineering and can be interested to the readers of this journal.

-         After careful revision of this paper, I advise to be revised with

minor revisions.

Comments on the present manuscript:

1-    Microstructure survey testing (SEM or AFM) is required for the SS-CFSST composite column before and after load stress trial.

2-    Authors stated that "Eight specimens were designed for this test, four with steel hoops and four with welding". Why did authors use 8 specimens four testing?  Based on what? International standard or referenced work?

3-    Incorporate in a table the mean differences in the load measurement values between hoop-type specimens and weld-type stub columns and what is recommended for civil engineering applications.

4-    Please, include the application economics of the new applied system (SS-CFSST) composite column, for civil engineering and the future exploitation capability in construction field.  

Author Response

(The authors gave the same response as above.)

Round 2

Reviewer 1 Report

The authors reply to reviewer comments in a separate document, but they do not incorporate any of the recommendations in the paper, even though a procedure was suggested to address those problems in an easy way.

The paper conclusions are weak and do not contribute much in creating new knowledge in the field od study.

Reviewer 3 Report

The authors provided all necessary corrections to improve the quality of the manuscript. I recommend accepting the manuscript for publication.